# Influence of Climatic Region and Feedstuff Type on the Co-Occurrence and Contamination Profiles of 54 Mycotoxins in European Grains and Forages: A Seven-Year Survey

**DOI:** 10.3390/toxins18010005

**Published:** 2025-12-20

**Authors:** Alexandra C. Weaver, Daniel M. Weaver, Luiz V. F. M. de Carvalho, Alexandros Yiannikouris

**Affiliations:** 1Alltech Inc., Nicholasville, KY 40356, USA; ayiannikouris@alltech.com; 2Independent Researcher, Orrington, ME 04474, USA; 3Alltech do Brazil, Maringá 87030-405, PR, Brazil; lcarvalho@alltech.com

**Keywords:** animal health, climate, co-occurrence, Europe, forages, grains, mycotoxins

## Abstract

Mycotoxins are global contaminants of feedstuffs and feeds that are linked to animal health and performance challenges and subsequently lead to economic burden. Negative effects of mycotoxin consumption may increase as a result of multiple mycotoxin co-occurrences. To assess mycotoxin challenge in Europe, a seven-year survey (2018 to 2024) of 1867 samples of grains (barley, maize, and wheat) and 818 forages (maize silage and grass silage) was conducted to assess the simultaneous presence of 54 mycotoxins using ultra-pressure liquid chromatography–tandem mass spectrometry. Results were categorized by feedstuff, harvest year, and climatic region to gain insight on mycotoxin occurrence, concentration and co-occurrence. Grains contained a mean 3.6 to 6.7 mycotoxin types per sample, while silages contained 3.1 to 6.0. Barley in the Nordic climate region had some of the highest *Fusarium* mycotoxin concentrations, while maize silage had consistently higher mycotoxin concentrations across all climate regions. The B trichothecenes and emerging mycotoxins had the highest rates of co-occurrence (52.4% to 74.2% of samples) in grains and maize silage. Co-occurrence data can serve as an initial framework for identifying or reasserting known environmental conditions that favor mycotoxin biosynthesis in distinct fungal taxa and for refining risk assessment of animals simultaneously exposed to multiple mycotoxins. Collectively, this survey shows that mycotoxin contamination and co-occurrence in grains and silages from Europe is expected, with differences occurring by feedstuff type and climatic region.

## 1. Introduction

Mycotoxins are naturally occurring small-molecule secondary molecules produced by certain mold species, primarily from *Alternaria*, *Aspergillus*, *Claviceps*, *Fusarium*, and *Penicillium* genera, among others [1]. The production of mycotoxins can occur globally in numerous crops and under a wide range of environmental and agronomic conditions, from field production through harvest, storage, and feed out. Some of the major factors affecting mycotoxin production include temperature, humidity, water activity, pH, aerobic conditions, fungal strain and microbial competition, substrate type and accessibility, and pest pressure [2]. Generally, mycotoxins may be placed into the categories of regulated, masked, emerging, and storage mycotoxins [3,4,5,6], although these are not mutually exclusive. Regulated mycotoxins are those that are typically well-known and have action or guidance levels globally. Masked mycotoxins are those that have been biologically modified by plant defense mechanisms and sequestered following Phase-II and Phase-III metabolism [5] and are more difficult to detect. Many emerging mycotoxins are produced by *Fusarium* molds and historically escaped detection by conventional analytical techniques, although advanced methods are allowing for more routine detection and quantification [3,7]. Storage mycotoxins are primarily produced by the *Penicillium* or *Aspergillus* taxa and typically develop during feedstuff or feed storage [4]. A few storage mycotoxins have regulatory guidelines in certain countries, but many mycotoxins in this group are not regulated.

Mycotoxins can be produced individually or in combination, where a single mycotoxin may be synthesized by multiple fungal species, a single mold can produce several distinct mycotoxins, and multiple different molds can be present on the same crop material [1]. In the United States, it is reported that maize grain and maize silage may contain 4.8 to 5.2 different mycotoxins per sample, with some of the emerging and masked mycotoxins having higher occurrence rates than the typical regulated mycotoxins [8]. In Europe, grains have been reported to contain up to 17 simultaneous mycotoxins [6]. A survey from Asian countries that assessed the presence of 25 mycotoxins also showed high prevalence of multiple mycotoxin contamination (including emerging mycotoxins) in feedstuffs, with the majority of rice bran samples containing 7 mycotoxins (49.6%) and maize containing 5 mycotoxins (40.8%) [3]. These survey results reflect the conclusion that multiple mycotoxin contamination is a common pattern globally in a variety of feedstuff types.

The consumption of individual mycotoxins by animals has been linked to a range of health and performance challenges. However, multiple mycotoxin mixtures can further increase the response due to heightened negative effects on intestinal health, immunity, and performance [9]. Together, these negative effects pose a substantial economic burden to producers. A previous meta-analysis [10] suggested that mycotoxin consumption in broilers could reduce the European Poultry Efficiency Factor by 22.5%, potentially resulting in an income loss of over EUR 1500 per flock. This estimate considered only reduced performance and increased mortality, excluding additional costs such as veterinary care. Due to the negative effects not only on animal health but also farm productivity and profitability, it is critical to understand the occurrence and co-occurrence of mycotoxins in animal feedstuffs to improve monitoring and mitigation programs. This study aimed to investigate prevalence, concentrations, and co-occurrence of 54 mycotoxins (11 groups) from the regulated, masked, emerging, and storage categories over a seven-year period, while also investigating the influence of feedstuff type and climatic region of Europe on these contamination patterns. This assessment of multiple mycotoxin contamination in several different crops by climatic regions of Europe is unique and could give insight into those regions with greater risk. Furthermore, co-occurrence patterns of mycotoxins were assessed through probabilistic co-occurrence modeling, which is of particular importance due to the combined influence of mycotoxins on animal performance and health. The information from this survey could help tailor local mycotoxin monitoring and mitigation by showing the mycotoxins that may be of primary importance to include in analysis programs and the potential risk for animals. Collectively, this survey may support a growing framework of knowledge looking to assess and predict mycotoxin risk by feedstuff type, growing region, and yearly environmental variability.

## 2. Results

### 2.1. Mycotoxin Contamination

There were some samples analyzed that had all mycotoxins below the detection limits, while the highest number was 18 mycotoxins simultaneously detected (Figure 1). Maize-based feedstuffs, including both grain and silage, had the highest mean number of mycotoxins per sample at 6.68 and 6.03, respectively. Wheat and grass silage had the lowest number of mycotoxins at means of 3.58 and 3.07 mycotoxins per sample. Furthermore, grass silage had the lowest maximum number of mycotoxins detected at 12 mycotoxins. The occurrence rate for the number of samples with different mean numbers of mycotoxins showed varying patterns by feedstuff type (Figure 2). A higher portion of wheat, barley, and grass silage samples had two or three mycotoxins, while the peak for maize grain and silage is greater at six to seven mycotoxins per sample.

When assessed by European climatic region, barley from the Nordic region had the highest mean number of different mycotoxins (9.33), which was significantly greater (*p* < 0.001) than barley from other regions (Figure 3). For maize grain, samples from the Oceanic region had a higher (8.17 versus 6.53, *p* = 0.017) mean number of mycotoxin than those from the Mediterranean region. A comparable regional difference was observed for wheat, albeit with lower counts, with the Oceanic region averaging 3.90 compared to 2.88 mycotoxins per sample for the Mediterranean samples (*p* = 0.021). Maize silage showed fewer regional differences, although samples from the Continental region had a higher mean number of mycotoxins than the Oceanic samples (6.54 versus 5.23; *p* < 0.001). Finally, grass silage samples from the Nordic region had a higher mean of 4.60 mycotoxins per sample than those from the Continental or Oceanic regions (3.25 and 2.88; *p* < 0.05). There were no maize, wheat, or maize silage samples from the Nordic region, and only one grass silage from the Mediterranean region, so impact of these regions within feedstuff could not be determined.

Among individual mycotoxins (Table 1), both the type of mycotoxin and the type of feedstuff influenced the proportion of samples exceeding the limit of detection (LOD), as well as the measured concentrations above the limit of quantification (LOQ). The most prevalent mycotoxins in barley (Table 2) were enniatin (ENN) A/A1 and ENN B/B1 at 75% of samples containing these mycotoxins above LOD, followed by deoxynivalenol (DON) at 64%. The mean DON level was 224 µg/kg, with a maximum of 27,089 µg/kg. For wheat (Table 2), the highest occurring mycotoxins were also ENN A/A1, ENN B/B1, and DON, ranging from 62 to 65%. The mean DON level was 168 µg/kg, with a maximum of 6758 µg/kg. Maize (Table 3) had a number of highly occurring mycotoxins. The highest occurrence rate was for fusaric acid (FA) at 93%, followed by the fumonisins (FUMs) at 70 to 80% and moniliformin (MON) at 77%. Fumonisin B1 (FB1) had the highest mean of the three FUMs at 1689 µg/kg and a maximum of 25,004 µg/kg. Only 57% of the samples contained DON above LOD. In contrast, DON was the most prevalent mycotoxin in maize silage (Table 4) at an occurrence rate of 97%, a mean of 1493 µg/kg, and a maximum of 72,362 µg/kg. Additionally, 89% of maize silage samples also contained FA. The two most prevalent mycotoxins in grass silage (Table 4) were FA at 59% and penicillic acid (PA) at 51%, which had a mean of 247 µg/kg and a maximum of 3946 µg/kg.

### 2.2. Mycotoxin Concentrations by Region

Concentrations of selected mycotoxin groups, which were most strongly influenced by climatic region, included B trichothecenes, A trichothecenes, FUMs, and emerging mycotoxins. Complete results for all mycotoxin groups analyzed for each feedstuff are provided in the Appendix A. To summarize, significant (*p* < 0.05) differences were observed between regions for barley for B trichothecenes, A trichothecenes, zearalenone (ZEA), and emerging mycotoxins. For maize, significant differences were observed for B trichothecenes, A trichothecenes, and FUMs, while for wheat, the differences were for B trichothecenes, FUMs, and emerging mycotoxins. Maise silage had differences by region for aflatoxins (AFs), A trichothecenes, FUMs, and FA, and grass silage only had significant differences by region for emerging mycotoxins and *Penicillium* mycotoxins.

Detailed results for two of the mycotoxin groups commonly showing regional differences, B trichothecenes, and emerging mycotoxins are shown in Table 5 and Table 6. Significant differences among regions were observed for B trichothecenes (Table 5) for the three analyzed grains, whereas regional differences were not observed for either of the silages. The highest median (1476.99 µg/kg), mean (3783.15 µg/kg), and maximum (28,987.66 µg/kg) concentrations of B trichothecenes were from barley in the Nordic region (*p* < 0.005). For maize grain, the highest mean (868.07 µg/kg, *p* < 0.005) was in the Oceanic region, and the highest maximum concentration (3974.41 µg/kg) sample was from the Mediterranean region. The B trichothecene concentrations for wheat were consistently lower than the other grains but did result in variation (*p* = 0.003) across regions, with the highest average (262.14 µg/kg) and maximum (7281.83 µg/kg) from the Continental region.

Emerging mycotoxin concentrations showed several significant variations (Table 6). Barley samples had differences by region (*p* < 0.005), with the highest levels of these mycotoxins in the Nordic region with a mean of 1560.89 µg/kg and maximum of 5144.72 µg/kg. Conversely, wheat showed the highest mean (*p* = 0.007) and maximum values for emerging mycotoxins in the Oceanic region at 67.41 µg/kg and 3114.48 µg/kg, respectively. Grass silage samples showed regional variations for emerging mycotoxins, with greatest mean of 129.11 µg/kg (*p* < 0.005) from the Mediterranean samples and maximum of 526.16 µg/kg detected in the Continental region.

### 2.3. Mycotoxin Concentrations by Year

Yearly variations were detected for both mycotoxin group and feedstuffs. Concentrations of selected mycotoxin groups that were largely influenced by year are shown in Figure 4 and Figure 5. Complete results for all mycotoxin groups and feedstuffs for each year are provided in the Appendix A. Results for grains showed several year-to-year differences in mean concentrations of mycotoxins. For barley, concentrations of AFs, ochratoxins/citrinin (OTs/CIT), B trichothecenes, A trichothecenes, FUMs, ZEA, emerging mycotoxins, and other *Aspergillus* mycotoxins showed significant (*p* < 0.05) variation by year (Figure 4, Appendix A). Similar mycotoxins were also influenced yearly by wheat, with significant (*p* < 0.05) variation found for AFs, B trichothecenes, A trichothecenes, FUMs, and emerging mycotoxins. For both barley and wheat, mean concentrations of B trichothecenes were highest in 2019 and lowest in 2022. Alternatively, both A trichothecenes and emerging mycotoxins in barley and wheat showed a general upward trend in average levels over time, from 2018 to 2024. Maize grain showed less yearly influence on mycotoxin concentrations, with significant (*p* < 0.05) variation by year detected only for AFs, FA, and emerging mycotoxins.

Forages also showed an influence of year on mycotoxin concentrations (Figure 5, Appendix A). For maize silage, significant (*p* < 0.05) differences were observed for AFs, A trichothecenes, FUMs, ZEA, and emerging mycotoxins. Additionally, AFs, B trichothecenes, FA, emerging mycotoxins, and ergot alkaloids all showed a significant (*p* < 0.05) effect of the year in grass silage. The group containing *Penicillium* mycotoxins was the only group not influenced by year for any feedstuff type.

### 2.4. Co-Occurrence of Mycotoxins

The co-occurrence of mycotoxin group pairs was assessed for each feedstuff. Across feedstuffs, there were numerous random associations between mycotoxin group pairs, particularly notable for the two silage matrices. For grains, the significant non-random associations were primarily positive. Barley samples had several co-occurrence interactions observed (14 positive and 5 negative), with the highest probability of co-occurrence above random association (*p* < 0.05, Table 7) between A trichothecenes and emerging mycotoxins at 0.330. The most frequently detected group pair for barley was B trichothecenes and emerging mycotoxins with a probability of 0.609, although this pair was determined to have random association. Probabilistic assessment for maize grain resulted in 11 positive and 2 negative co-occurrences. The significant (*p* < 0.05) co-occurrences, at rates above random association, were between B trichothecenes and FA (0.559), FUMs and FA (0.755), and FUMs and emerging mycotoxins (0.709). However, the most highly co-occurring pair was FA and emerging mycotoxins at 0.808, although this was non-significant (random association). For wheat, only positive significant co-occurrences were found, with the highest co-occurrence of 0.529 between B trichothecenes and emerging mycotoxins.

Assessment of pair attributes showed that barley and wheat had the greatest percentage of mycotoxins contributing to positive co-occurrences. For barley (Figure 6a), 40–50% of ZEA, A trichothecenes, and FA contributed to positive co-occurrences with the remaining being random. The B trichothecenes showed predominantly positive interactions (50%) but also demonstrated negative (20%) and random (30%) associations. For wheat (Figure 6c), regardless of the mycotoxin category, the mycotoxin pairings contributed solely to positive (20% to 70%) or random co-occurrences. A high percentage of positive pairings contained ZEA (71%), as well as FA (50%) and emerging mycotoxin (40%). Maize (Figure 6b) only had 22% of the positive pairs containing B trichothecenes, while most groups are contributing to the random mycotoxin co-occurrence.

Maize silage had a lower number of significant associations (Table 8), with more negative (five pairs) than positive (four pairs) associations and a greater number of random or non-significant pairs (Table 8, Figure 6d). The B trichothecenes and FA at 0.863 (*p* < 0.05) had the highest rate of occurrence greater than the null. Other high but non-significant random co-occurrences were for B trichothecenes and emerging mycotoxins (0.742) and FA and emerging mycotoxins (0.681). Maize silage had a low number of both positive and negative mycotoxin interactions and more random associations. Although highly co-occurring, only 22.2% of B trichothecene and 10% of FA interactions contributed toward positive co-occurrences out of all interactions. Across all groups, most pairings (81.2%) followed random chance co-occurrence.

Grass silage showed a similar trend for a greater number of negatively associated co-occurrences (six pairs) than positive (three pairs) and a majority of random co-occurrences (Table 8). Across all groups, grass silage had the highest percentage of negatively associated pairings at 14.8% (Figure 6e). The B trichothecenes and emerging mycotoxins occurred less than expected (0.288). Although at lower probability rates, notable positive mycotoxin pairs were FA and emerging mycotoxins (0.316) and FA and *Penicillium* mycotoxins (0.321).

## 3. Discussion

Mycotoxin contamination of feeding materials for animals is common worldwide and may alter the quality of the commodity as well as the health, performance, and profitability of the animals consuming this material. Studies have shown that mycotoxin contamination can also contribute to raising the carbon footprint of production [10,11] through altered feed efficiencies and increased mortality rates. Furthermore, when considering the co-contamination of multiple mycotoxins, this could raise the risk to animal health and performance. While mycotoxin contamination of feedstuffs is the norm, as shown in both the current survey and previously published surveys [8,12,13], the presence and concentrations of mycotoxins are also thought to be associated in part with climate change and altered agricultural practices [14].

Mycotoxin content can vary by matrix type and is influenced by the European climatic region in which the crop was grown and stored. Across feedstuffs, maize grain contained the greatest mean number of mycotoxins, with 6.68 different mycotoxins per sample, followed by maize silage at 6.03. Maize is considered to be one of the most susceptible crops to pathogenic fungi [14]. Interestingly, while maize products had the highest average number of mycotoxins across all samples and typically contained the highest detected maximums, this survey showed the prevalence of important regional variations. For example, barley in the Nordic region had the highest mean number of mycotoxins overall at 9.33, even though the mean number for barley across all regions combined was only 4.53 mycotoxins. Furthermore, barley was one of the feedstuffs, with the greatest number of mycotoxin groups (four groups) that significantly differed by region and had some of the highest average concentrations of mycotoxins within these groups. Within the Nordic region, all samples in this survey were from Finland. Barley is one of the most important crops produced in Finland, representing about one-quarter of the cultivated area [15]. As such, the high prevalence and concentrations of mycotoxins in this feedstuff from this region could have significant health and economic effects.

Significant yearly variations in mycotoxin content were observed across each feedstuff type. The small grains, barley and wheat, both showed yearly differences across a number of mycotoxin groups. In general, barley showed the highest mycotoxin concentrations in 2020 and 2023, while wheat results showed elevated concentrations of several mycotoxin groups in 2020 and 2024. Interestingly, both barley and wheat showed no difference in FA levels across years, with low concentrations and occurrence rates less than 10%. Maize, on the other hand, only had yearly differences for FA and emerging mycotoxins, with concentrations being highest in 2021 and 2022, respectively. Maize also contained a high occurrence rate of FA, with greater than 92% of the samples containing this mycotoxin, while occurrence rates of the various emerging mycotoxins were low. Maize silage samples were also impacted by year, with 2018 and 2023 generally having the highest concentrations, although yearly differences were not observed for B trichothecenes and FA, despite these being the two most prevalent mycotoxins. As such, it may be concluded that these two mycotoxin groups may be of constant presence in maize silage. More similar to the small grains, grass silage samples showed several yearly differences, with 2020 having several higher mycotoxin concentrations.

When looking across years, feedstuffs, and mycotoxins, it may be summarized that 2020 and 2023 were among the years that showed the highest mycotoxin concentrations, particularly for the small grains and grass silage. Europe is reported to be the fastest warming continent [16]. The Copernicus Climate Change Service report [16] for Europe states that 2024 was the warmest year on record since 1900, followed by 2020 and 2023. Furthermore, in 2024, 34% of the land area experienced above-average precipitation, while in 2023, about 7% of the land area experienced above-average precipitation. Although the results from this mycotoxin survey have not been correlated with any statistical model, it may be suggested that feedstuffs such as barley, wheat, and grass silage could have increased mycotoxin occurrence during years of elevated temperature stress. In contrast, mycotoxin contamination of maize and maize silage may be promoted by different weather patterns, notably the slightly lower (although still above-average) temperatures observed in years other than 2020, 2023, and 2024 [16]. In addition to temperature and precipitation amount, it should also be remembered that the timing of these weather events during plant and grain development is also critical [17]. Although the year 2025 is not complete at the time of this survey, reports indicate that 2025 had the fourth warmest summer (June to August) for the European continent, with pronounced variation in precipitation among regions [18]. As such, it may be expected that mycotoxin risk will continue to increase. Although the conclusions from this survey regarding mycotoxin variation by climatic region and year are largely speculative due to a lack of detailed and local weather (temperature, precipitation) data, these results do provide a starting point for concepts to investigate. Further research is needed to link location data, mycotoxin results, and weather patterns to have a more accurate understanding of climate influence and mycotoxin predictions.

Identifying patterns of co-occurrence for groups of mycotoxins may spur further research and insights. Co-occurrence is investigated under a null (random) model that tests if there are combinations of pairs that show significant positive or negative associations [19]. These associations, or patterns of species occurrence, can provide insight into the pair’s relationships within the environment, as they relate to habitat, competition, symbiotic relationships, etc. Positive associations represent the co-occurrence of pairs that appear greater than random chance, suggesting that a pair of mycotoxin groups co-occurred together more than expected in that environment. This positive co-occurrence may be due to a similarity in mold species producing those mycotoxins, a mold producing more than one mycotoxin, or similar environmental stimuli promoting the production of varying mycotoxin groups, among other factors. On the other hand, negative associations are those where species occur less often than expected by random chance, often due to species competition, avoidance, or sequential matrix-associated evolution [19]. In relation to mycotoxins, this may occur among molds that require differing environmental conditions to grow or produce mycotoxins [20]. Further investigation of species attributes can provide information on the contribution of the individual group to the positive, negative, or random associations, and indicate whether the co-occurrences are evenly distributed among groups rather than clustered within a few groups [21]. With this data, results may give insights into the types of mycotoxins expected to be found on each commodity, as well as the likelihood of mycotoxins coming from *Fusarium*, *Penicillium*, *Aspergillus*, or *Claviceps* species.

The results obtained in this research showed that there were more significant deviations from random expectations for grains than for silages, with grains having mostly positive deviations (40 positives versus 7 negatives). Across all grains and maize silage, the highest probabilities of co-occurrence (both positive and non-significant) were between mycotoxin groups produced primarily by *Fusarium* molds. Notably, FA and emerging mycotoxins were highly co-occurring with B trichothecenes. *Fusarium* represents a large group of mold species, which produce numerous mycotoxins including B and A trichothecenes, FUMs, ZEA, FA, beauvericin, ENN, and MON [14]. Furthermore, some mycotoxins such as FA or MON are reported to be produced by more than 12 and 30 *Fusarium* species, respectively [22,23]. In this survey, there were high positive co-occurrence rates between B trichothecenes and FA in maize and maize silage, with 55.9% and 86.3%, respectively, indicating that these two mycotoxin groups are not only common in maize-based commodities but also may be promoted by similar environmental conditions. Maize and maize silage also contained high co-occurrence of B trichothecenes and emerging mycotoxins (52.4% and 74.2%) and FA and emerging mycotoxins (80.8% and 68.1%), although these were associated with the models’ random expectations. Interestingly, although B trichothecenes are a highly co-occurring group, they only contributed to 22% of all positive pairs, showing that they are highly occurring but at rates expected by the models. Across previously published surveys, mycotoxin members of the B trichothecenes are found to be highly occurring [6,24].

It is interesting to note that AFs had low occurrence rates across matrix types, at rates of less than 2.7% of samples having AFs above LOD, with the exception of maize that had the occurrence of AFs up to 13%. At the same time, co-occurrences of AFs with other mycotoxins were also low, with an observed maximum (random association) between AFs and FA in maize at 12.7%. Furthermore, analysis of AFs by year was marked by only certain years, with elevated concentrations across feedstuffs. The lower occurrence rates of AFs may reflect a more sporadic production pattern, with episodic spikes driven by periods of elevated temperatures and water activity (a_w_)—conditions that have recently become more common in parts of Europe [2,16]. Due to the low detection frequency of AFs, establishing robust co-occurrence patterns has been more challenging. Nevertheless, these toxins remain highly relevant as they constitute the only mycotoxin category with enforced regulatory limits. In this survey, levels of AFs were detected above permissible thresholds in some cases, in particular for maize and maize silage (maximum values of 451 µg/kg and 152 µg/kg).

Emerging mycotoxins play an important role in observed mycotoxin co-occurrences. In barley, the highest co-occurrence was for B trichothecenes and emerging mycotoxins at 60.9% (positive), showing that these mycotoxins often occur together and at rates above random chance. Previous surveys have also found high rates of emerging mycotoxins in barley [25]. Interestingly, although emerging mycotoxins had a high co-occurrence with B trichothecenes, this mycotoxin group contributed only 10% to all positive co-occurrences, indicating that in the majority of interactions, these mycotoxins were at expected rates. Wheat was similar to barley regarding co-occurrences, with the highest co-occurrences between B trichothecenes and emerging mycotoxins at 52.9% (positive). Wheat was similar to barley regarding co-occurrences, with the highest co-occurrences between B trichothecenes and emerging mycotoxins at 52.9% (positive).

The key emerging mycotoxins detected in barley and wheat, as well as maize silage, were ENN A/A1 and ENN B/B1 at 54 to 75% occurrence rates. In contrast, maize grain had a higher occurrence of MON than ENN. There are numerous mycotoxins that fall into this ‘emerging’ category, including, for example, several produced by *Alternaria* species such as tenuazonic acid, alternariol monomethyl ether, and altertoxin [26]. Although the current survey only included one *Alternaria* toxin (alternariol), it covered many of the key emerging mycotoxins (such as ENN A/A1, ENN B/B1, and MON). These emerging mycotoxins can be produced by multiple mold species, but it is reported that ENN is most favorably produced at a_w_ and temperature combinations of a_w_ 0.994/25 °C, while MON production occurs more readily at lower moisture levels of a_w_ 0.960/25 °C [27]. The production of DON by *F. graminearum* is also similar at a_w_ 0.960/25 °C, supporting the high co-occurrence of these mycotoxins. Due to these high rates of co-occurrence, the potential for feedstuffs to be contaminated with multiple mycotoxins may be considered the norm. For the animal industry, mycotoxin co-contamination is of importance as the consumption of multiple mycotoxins increases the total risk to cause greater negative effects on health or performance through additive or potential synergistic relationships [9,28,29,30]. Pertaining specifically to the interactions of DON with ENN, MON, or FA, there are a limited number of references, with those that are available cover only a few animal species and lack investigation with naturally contaminated feed materials. Generally, trials investigating DON with ENN or MON report neutral (non-related) or additive-type interactions [31,32]. In contrast, the presence of DON with FA is considered to vary widely between neutral, additive, and synergistic [33,34]. Given their high co-occurrence, further research on the combined toxicity of these mycotoxins under natural contamination scenarios and across different animal species is warranted. Such work could help explain discrepancies observed among naturally contaminated matrices containing unaccounted mycotoxins (e.g., FA) and controlled, single-toxin dietary challenge studies.

In contrast to the cereal grain-based commodities, grass silage had a pattern of few positive and more negative associations, which may indicate that grass has a pattern of mold growth differing from that of other commodities, in part due to the grass substrate composition, growing conditions, harvest timing, and storage management. Generally, grass silage had lower co-occurrence rates of mycotoxins than the other feedstuff types, shown, for example, with the co-occurrence rate for B trichothecenes and emerging mycotoxins at only 28.8% and having a negative relationship. This negative association indicates that these two groups are occurring less than model expectations or less than observed by random chance in nature. In fact, many of the *Fusarium* mycotoxins occurred less in grass silage than expected. These results may be supported by previous research showing that grass silage contained fewer fungal metabolites (mean 20; range 12 to 27) out of 106 analyzed, whereas maize silage contained a mean of 26 (range 19 to 64) fungal metabolites [35].

There are several unknown factors related to the growth and production of grass silage samples assessed in this survey. As such, some level of caution may be warranted when considering the variability and traceability of mycotoxin risk in grass silage (and similarly maize silage) in this longitudinal survey. For example, it is unknown whether the cut number or harvest timing was consistent across grass silage samples. It is also unknown as to which species of grass were used for each silage sample. Grass species used to make this forage source can vary widely across Europe, which may influence the associated susceptibility to different types of molds, although this has not been statistically confirmed [36]. Moreover, the duration and conditions of storage before analysis, which can markedly influence both the total mycotoxin load and the pattern of co-occurrence present at the time of consumption, were not captured.

Previous research has demonstrated that the timing of cutting and storage conditions may have the greatest influence on mycotoxin (such as DON, ZEA, and T2) content in grass silage [36]. For example, DON content of grass harvested in the Czech Republic was shown to be lower in June than from late July to October, although it should be noted that overall each cutting had relatively low concentrations of DON with a maximum mean of 51.9 µg/kg [36]. As such, harvest times could have played a role in the deviation from the expected occurrence of B trichothecenes in the current survey, but it also may be concluded that these mycotoxins may in fact be occurring less and at lower concentrations in this commodity than others [35]. Perhaps of more importance is the effect of ensilaging, which was shown by [36] to increase DON content by about 408% (up to 167.7 µg/kg). Alternatively, in the current survey, the highest co-occurrence rate of 32.1% was found for FA and the *Penicillium* mycotoxins group (positive), which was much greater than for other matrix types. This high rate of *Penicilliums* in grass silages is supported by previous work [4,35]. *Penicillium* molds typically favor growth and production of mycotoxins on crops post-harvest, being able to grow under a wide range of environmental conditions that include lower a_w_ (0.79–0.83), pH (3.0–6.0), and oxygen (1%) levels [37]. However, these molds can also grow and produce mycotoxins on crops pre-harvest when wet conditions occur in the field [38]. The majority of grass silage samples originated in the Oceanic and northern Continental regions, where grass silage is typically collected over several cuttings from May to August [39,40]. Due to this wide range of harvest time and processing, this commodity could be exposed to a variety of environmental conditions. Furthermore, grass silage must undergo proper fermentation for best storage quality. Silage that has lower dry matter and slower fermentation has been correlated with increased mycotoxin risk [41]. Due to the difference in production and the influence of both field and storage, this feedstuff may have a different mycotoxin profile from the cereal-based commodities.

## 4. Conclusions

This survey of European grains and forages provides insight into mycotoxin concentrations, occurrence, and co-occurrence between 2018 and 2024. Although it represents only a snapshot of a mycotoxin surveillance program, this survey can be used to understand potential future risk of mycotoxin contamination in animal feedstuffs and ingredient commodities. Furthermore, this survey provides novel insight into mycotoxin profiles according to the European climatic region and mycotoxin co-occurrence patterns for a wide range of mycotoxins including those in the emerging mycotoxin and storage mycotoxin categories.

It has now been established that it is the norm rather than an exception for grains and forages to contain multiple mycotoxins. Maize contained the greatest diversity of mycotoxin, averaging 6.68 distinct toxins per sample, while grass silage contained the lowest average at 3.07. Climatic region played an important role in the observed number of mycotoxins and mycotoxin concentrations. Surprisingly, barley from the Nordic region was one of the more contaminated feedstuffs, with an average of 9.33 mycotoxins per sample and the highest mean and maximum concentrations for several mycotoxin groups including B trichothecenes and emerging mycotoxins. Harvest year also played a role in observed mycotoxin content. In general, small grains and grass silage appeared to have greater mycotoxin risk in 2020, 2023, and 2024, which were years marked by higher average temperatures across Europe, along with elevated rainfall in some regions. Both barley and wheat had high co-occurrence of B trichothecenes and emerging mycotoxins. However, grass silage had the highest co-occurrence of *Penicillium* mycotoxins with FA. In contrast, maize and maize silage appeared to have different contamination patterns. Maize had the highest co-occurrence between FUMs and FA or emerging mycotoxins and appeared to have higher concentrations in 2021 and 2022. Maize silage had the highest co-occurrence of B trichothecenes and FA, with generally increased risk in 2018 and 2023. As such, it may be concluded that mycotoxin occurrence and content vary by feedstuff type, region, and year, which may be expected due to varied environmental conditions. Future research is needed to better model and predict the relationship between climate and mycotoxins.

The presence of multiple mycotoxins in different feedstuffs of European origin appears to be the norm and could increase the risk of mycotoxins to animal performance and health. Furthermore, the combination of several feedstuffs containing multiple mycotoxins can further increase risk and result in final mycotoxin intake well above the guidelines suggested by government or regulatory groups. The knowledge gained from this survey may help to support and advance mycotoxin monitoring and mitigation programs. For example, local mycotoxin testing may be tailored based upon those mycotoxins that more frequently occur in that climatic region or feedstuff, including not only the regulated but also masked, emerging, or storage mycotoxins. Maize-based commodities should be generally considered to have the greatest mycotoxin contamination, with the exception of barley from the Nordic climatic region. Furthermore, results demonstrated high mycotoxin co-occurrence, indicating a need for not only an increase in the number of mycotoxins routinely analyzed by individuals and regulatory groups, but also an awareness for a potential increase in negative effects on animals. The high co-occurrence rates of B trichothecenes with FA and emerging mycotoxins should be noted and may suggest a need for continued research aimed at understanding the effects of these mycotoxin groups together on animal performance and health under natural and common contamination levels. Overall, the detection, quantification, and assessment of multiple mycotoxins in feedstuffs should be considered a routine component of a mycotoxin monitoring program, with consideration of management strategies that reduce mycotoxin risk to and within the animal.

## 5. Materials and Methods

### 5.1. Sample Collection and Analysis

Samples were submitted from feed and animal production facilities as part of the annual Alltech European Harvest Analysis Survey, which focuses on new crop samples for each year. Samples of barley, maize, wheat, maize silage, and grass silage were collected from 23 countries across European climatic regions (Figure 7). Countries were divided into four climatic regions, Continental, Mediterranean, Nordic, or Oceanic, based on a classification similar to [42]. Samples were collected from new crops and represented 7 harvest years, from 2018 to 2024. Most samples were collected between July and December of each year. A total of 2635 samples were included in this survey (Table 9), comprising 710 barley, 387 maize, 743 wheat, 328 maize silage, and 467 grass silage samples.

Each sample was analyzed for 54 different mycotoxins, representing 11 groups (Table 1), by the Alltech 37+^®^ Analytical Services Laboratory located in Dunboyne, Ireland (International Organization for Standardization (ISO) and the International Electrotechnical Commission (IEC)17025:2005 No. 79481 and ISO/IEC 17025:2017 official accreditation No. 79481, Perry Johnson Laboratory Accreditation, Inc., Troy, MI, USA). Methods for proper sampling procedures are based on those described by the USDA Grain Inspection Handbook [43] and Undersander et al. [44]. Procedures outline the collection of samples to obtain a homogenous representation of the lot. It was recommended that each sample be packaged to preserve the integrity of the contents such as double bagging, vacuum sealing, and refrigeration at <4 °C. The Alltech 37+^®^ laboratory requests samples of 200–400 g and recommends that samples be submitted to the laboratory immediately after collection and sent via overnight shipping.

Mycotoxin quantification was completed using ultra-pressure liquid chromatography with tandem mass spectrometry (UPLC-MS/MS, Waters Acquity UPLC-TQD and XEVO TQ-S systems, Waters Corp., Milford, MA, USA), with an electrospray ionization interface working in positive and negative mode following the methods of Jackson et al. [45]. Laboratory processing is previously described by Weaver et al. [8]. Briefly, silage samples were freeze-dried to reduce moisture content, while grain sample moisture was determined by thermos gravimetry (Moisture Analyzer MB45, Ohaus Corp., Parsippany, NJ, USA) and processed as is. Each grain and silage sample was finely ground and placed in reaction tubes spiked with 20 µL of a 1:1:1 internal standard mixture ([^13^C_15_]-DON at 4.5 µg/g; [^13^C_18_]-ZEA at 4.5 µg/g; [^13^C_17_]-AFB1 at 273 ng/g and 20 µL of [^13^C_34_]-FUM at 2.3 µg/g mixture). Samples were extracted in an acetonitrile/water/formic acid mixture (84.0:15.9:0.1, *v*/*v*/*v*) for no more than 18 h at room temperature (RT), with shaking (250 rpm, New Brunswick Scientific, Enfield, CT, USA). All samples were centrifuged at 12,000 rpm for 10 min (Beckman Coulter Inc., Fullerton, CA, USA). Supernatant (400 µL for grains; 800 µL for silages) was collected, added to an autosampler vial, and dried under an N2 stream before being reconstituted in 400 µL of loading buffer (LB) consisting of water/acetonitrile/formic acid (95.0:4.9:0.1, *v*/*v*/*v*) containing 10 mmol L^−1^ ammonium acetate. Blank samples and a certified reference material were used alongside, and a system suitability check was performed before and after every 10 samples. The recovered levels of certified mycotoxins were accurate within ±25%.

### 5.2. Statistical Analysis

The average number (count) of mycotoxins detected in a sample was assessed among feedstuff and climatic region using ANOVA and gauged significance at p<0.05. These analyses were performed with R version 2024.12.1 [46] using built-in functions. The average number of toxins in a sample among climatic regions were statistically different (*p* < 0.05) for all feedstuffs. Therefore, further exploration of these differences was conducted using Tukey’s Honestly Significant Difference (HSD) post hoc pairwise comparisons. Tukey’s HSD adjusted the alpha critical value used to determine statistical significance, which reduced the probability of making a Type 1 error. These pair-wise tests allowed for examination of the differences among the average number of mycotoxins between each climatic region for each feedstuff. Mycotoxin concentration was assessed separately among climatic region and year for each feedstuff using analysis of variance (ANOVA). Statistically significant differences were identified when *p*-values were less than an alpha value of 0.05.

The probability of co-occurrence was calculated based on a grain or silage sample containing one mycotoxin group simultaneously with a second mycotoxin group. Assessment of mycotoxin co-occurrence was conducted using the “cooccur” package in R version 1.3 [21] and has been described previously by Weaver et al. [8]. To summarize, pairwise co-occurrences of mycotoxin groups were examined following the methods of Veech [47] and Griffith et al. [21]. Co-occurrence of mycotoxin group pairs was evaluated as the number of instances that samples contained two mycotoxin groups, relative to the expected number of times that these pairs would occur together under the null hypothesis of random chance. The construction of matrices for each analysis was completed using 0’s and 1’s. Samples having a positive, above LOQ concentration were assigned a “1”, and analyses that did not detect a mycotoxin (below LOQ), reported as a zero, were assigned a “0”. The probability of co-occurrence was calculated following that of Griffith et al. [21] to assess if a feedstuff would contain an individual mycotoxin given that it contained a second individual mycotoxin as,(1)Pj=Nxj×N−NxNy−jNNy
where *Nx* is the number of samples where mycotoxin *x* occurs, *Ny* is the number of samples where mycotoxin *y* occurs, and *N* is the total number of samples that were analyzed, where both *x* and *y* mycotoxins could occur and *j* = 1 to *Nx* samples. The number of ways of selecting *j* samples that have mycotoxin *x*, given that there are *Nx* samples Nxj, was analyzed. The expression N−NxNy−j represents the number of possibilities of selecting *Ny* − *j* samples that have mycotoxin *y* without mycotoxin *x* given. The number of combinations that *Ny* samples could be obtained out of all *N* samples  NNy was calculated. A co-occurrence matrix and an observed-expected plot were created to visually evaluate the probabilities of observed frequencies of co-occurrence. These were reported as positive (orange), negative (blue), or random (gray) in relation to the expected co-occurrences. Finally, to assess the contribution of each mycotoxin group toward the positive and negative pair associations, the R functions “pair.attributes” and “pair.profile” were used as described by Griffith et al. [21].

## Figures and Tables

**Figure 1 toxins-18-00005-f001:**
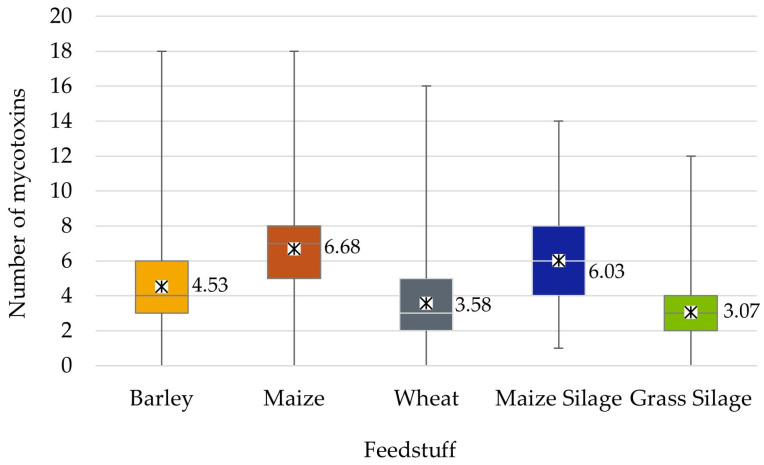
Box-and-whisker plot of the number of mycotoxins in different feedstuffs detected by ultra-pressure liquid chromatography–tandem mass spectrometry assessed over seven harvest years in Europe. Asterisks indicate the mean with data label included; horizontal lines within the boxes represent quartile 1 (lower line), median (middle line), and quartile 3 (upper line), and vertical lines represent the minimum and maximum number of mycotoxins detected in the samples.

**Figure 2 toxins-18-00005-f002:**
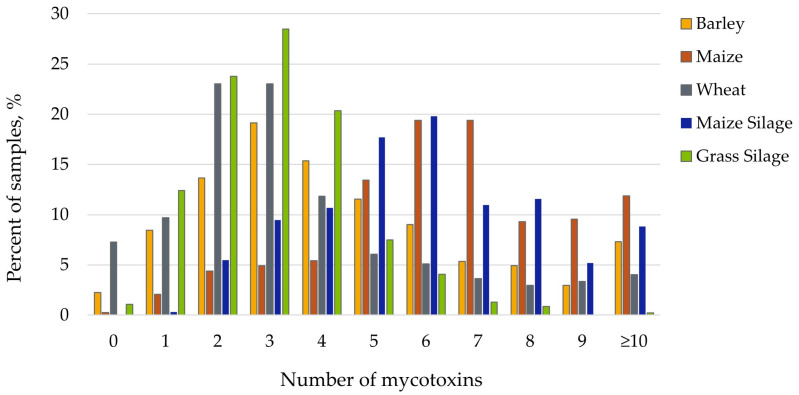
Percentage of samples (%) containing different numbers of mycotoxins out of a total of 54 detected by ultra-pressure liquid chromatography–tandem mass spectrometry assessed over seven harvest years in Europe. Feedstuffs include barley (yellow), maize (orange), wheat (gray), maize silage (blue), and grass silage (green).

**Figure 3 toxins-18-00005-f003:**
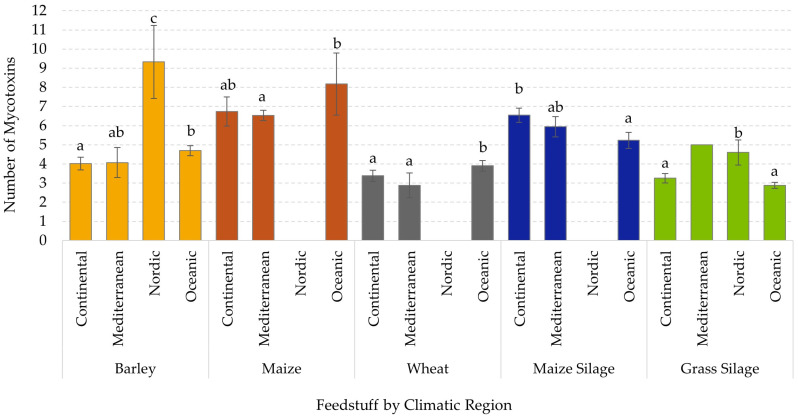
Influence of climatic region on the mean number of mycotoxins detected in different feedstuffs by ultra-pressure liquid chromatography–tandem mass spectrometry assessed over seven harvest years. Data bars represent mean and 95% confidence interval. Means not sharing a common letter for region within a feedstuff type differ significantly (*p* < 0.05). A lack of bars indicates no samples were available for that region.

**Figure 4 toxins-18-00005-f004:**
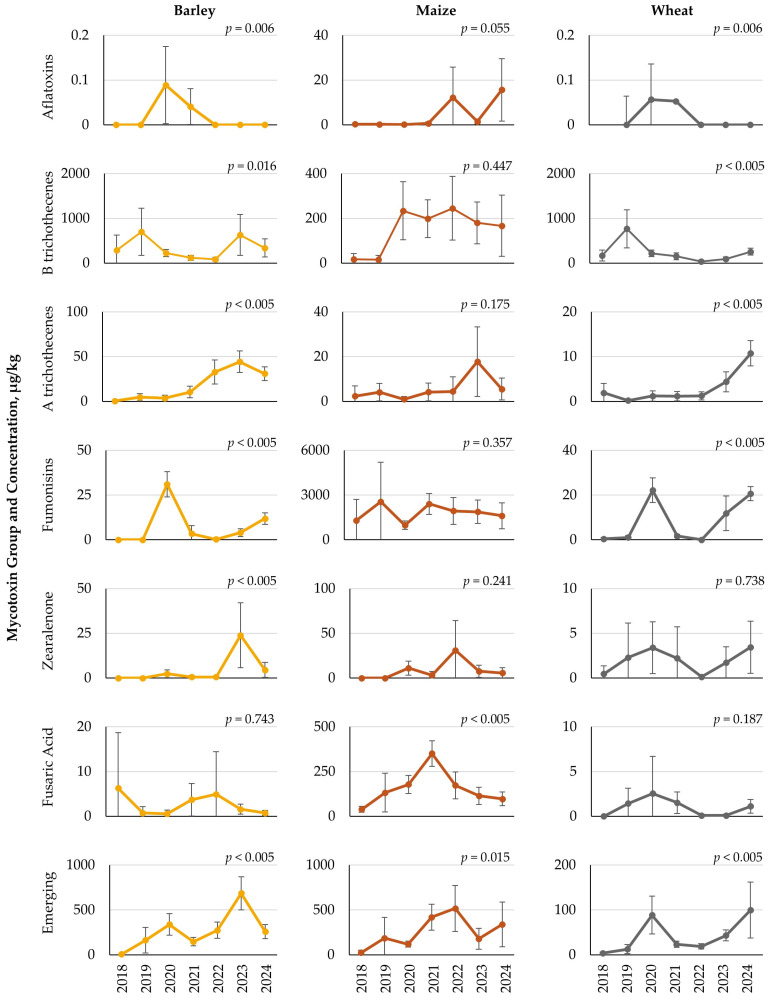
Yearly variation in mean mycotoxin concentrations (µg/kg) detected in barley, maize, and wheat grains from Europe, collected over seven harvest years, from July to December each year between 2018 and 2024. Data represents mean concentration (µg/kg), with error bars for the 95% confidence interval. Lower confidence intervals that generated negative values are shown as zero. *p*-values are reported for overall yearly variation by mycotoxin group within each grain type.

**Figure 5 toxins-18-00005-f005:**
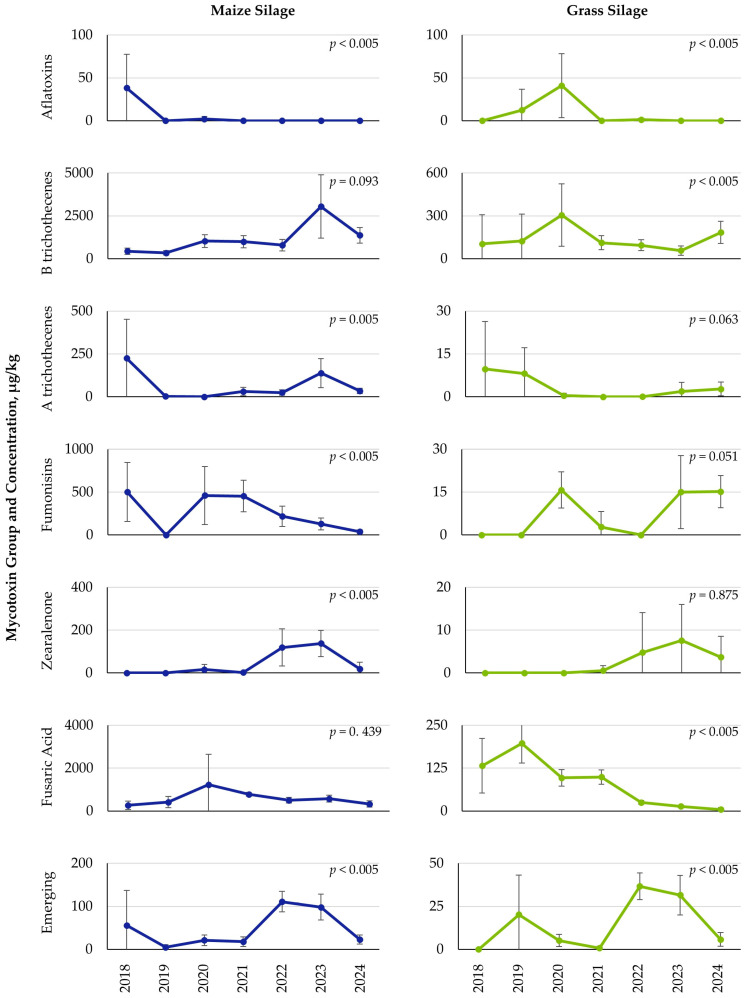
Yearly variation in mean mycotoxin concentrations (µg/kg) detected in maize silage and grass silage samples from Europe, collected over seven harvest years, from July to December each year between 2018 and 2024. Data represents mean concentration (µg/kg) with error bars for the 95% confidence interval. Lower confidence intervals that generated negative values are shown as zero. *p*-values are reported for overall yearly variation by mycotoxin group within each forage type.

**Figure 6 toxins-18-00005-f006:**
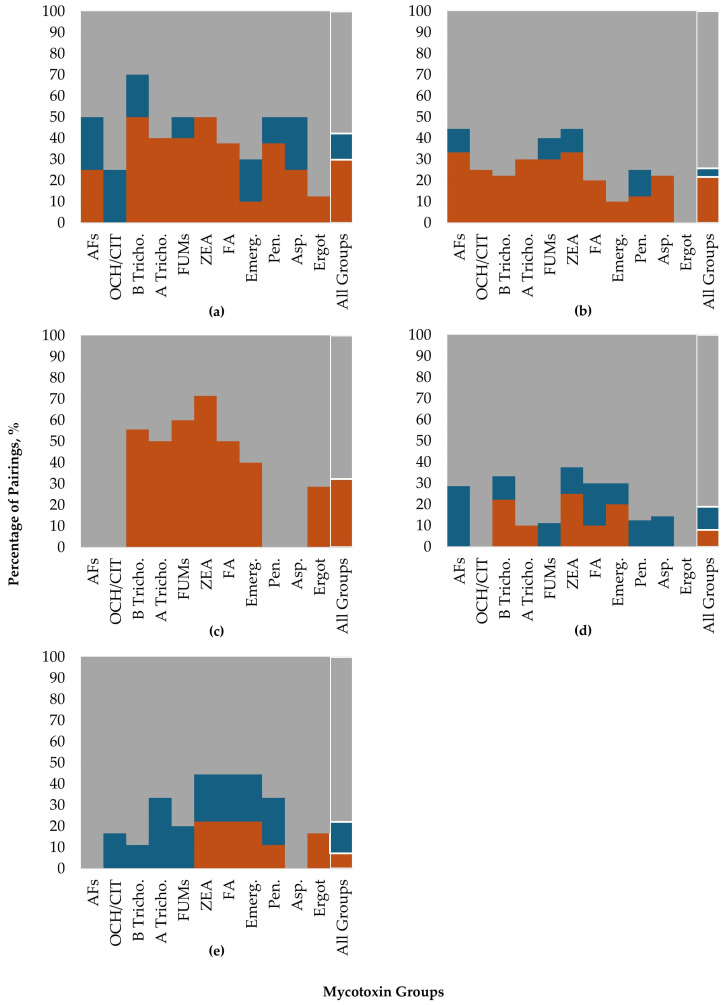
Pair attributes showing the contribution (%) of each mycotoxin group individually or all groups together toward the positive (orange), negative (blue), or random (gray) associations assessed by probabilistic co-occurrence modeling. Results are shown for barley (**a**), maize (**b**), wheat (**c**), maize silage (**d**), and grass silage (**e**). Mycotoxin groups include total aflatoxins (AFs), ochratoxin and citrinin (OCH/CIT), type B trichothecenes (B Tricho.), type A trichothecenes (A Tricho.), total fumonisins (FUMs), zearalenone (ZEA), fusaric acid (FA), emerging mycotoxins (Emerg.), other Penicillium mycotoxins (Pen.), other Aspergillus mycotoxins (Asp.), and ergot alkaloids (Ergot).

**Figure 7 toxins-18-00005-f007:**
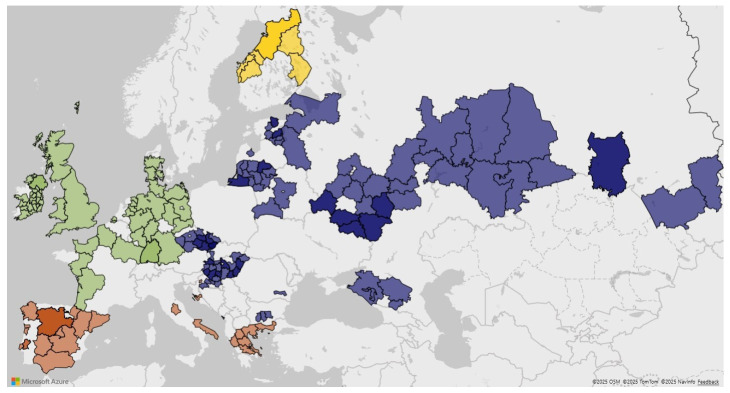
Regional locations of samples and climatic classification of European countries for Continental (blue), Mediterranean (orange), Nordic (yellow), and Oceanic (green). Countries without color did not have sample representation.

**Table 1 toxins-18-00005-t001:** Analyzed mycotoxins assessed in this survey by ultra-pressure liquid chromatography–tandem mass spectrometry (Alltech 37+^®^ Analytical Services Laboratory, Dunboyne, Ireland).

	Group	Mycotoxin	Abbreviation	Limit of Detection, µg/kg	Limit of Quantification, µg/kg
1	Aflatoxins	Aflatoxin B1	AFB1	0.130	0.429
2		Aflatoxin B2	AFB2	0.539	1.780
3		Aflatoxin G1	AFG1	0.145	0.480
4		Aflatoxin G2	AFG2	0.148	0.490
5	Ochratoxin	Ochratoxin A	OTA	2.076	6.850
6		Ochratoxin B	OTB	2.218	7.320
7		Citrinin	CIT	0.308	1.015
8	B Trichothecenes	Deoxynivalenol	DON	4.348	14.350
9		3-acetyl-deoxynivalenol	AcDON3	2.288	7.550
10		15-acetyl-deoxynivalenol	AcDON15	1.794	5.920
11		Deoxynivalenol-3-glucoside	DON3G	10.973	36.210
12		Nivalenol	NIV	49.918	164.730
13		Fusarenon X	FUSX	2.514	8.295
14	A Trichothecenes	T-2 toxin	T2	0.752	2.481
15		HT-2 toxin	HT2	3.838	12.665
16		Diacetoxyscirpenol	DAS	1.520	5.017
17		Neosolaniol	NEO	1.824	6.020
18	Fumonisins	Fumonisin B1	FB1	20.632	68.086
19		Fumonisin B2	FB2	1.822	6.012
20		Fumonisin B3	FB3	4.998	16.493
21	Zearalenone	Zearalenone	ZEA	2.570	8.482
22	Fusaric Acid	Fusaric acid	FA	2.376	7.840
23	Emerging	Beauvericin	BEA	0.470	1.550
24		Moniliformin	MON	1.586	5.233
25/26		Enniatin A/A1	ENN A/A1	0.030	0.090
27/28		Enniatin B/B1	ENN B/B1	0.070	0.220
29		Phomopsin A	PHOM A	0.040	0.130
30		Alternariol	AOL	1.393	4.598
31	*Penicilliums*	Patulin	PAT	16.837	55.562
32		Mycophenolic acid	MPA	1.021	3.370
33		Roquefortine C	ROQC	1.776	5.860
34		Penicillic acid	PA	7.427	24.510
35		Citreoviridin	CITR	2.572	8.487
36		Wortmannin	WORT	0.771	2.545
37	*Aspergillus*	Gliotoxin	GLIO	5.664	18.692
38		Sterigmatocystin	STMC	0.185	0.612
39		Cyclopiazonic Acid	CPA	0.989	3.265
40		Verruculogen	VERR	0.335	1.104
41/42	Ergot Alkaloids	Ergometrin(in)e	ERGMET	0.579	1.911
43/44		Ergotamin(in)e	ERGAM	0.507	1.673
45/46		Ergocristin(in)e	ERGCR	2.945	9.717
47/48		Ergosin(in)e	ERGSI	1.166	3.849
49/50		Ergocornin(in)e	ERGCO	0.838	2.764
51/52		Ergocryptin(in)e	ERGCYP	0.811	2.677
53		Lysergol	LYS	0.461	1.522
54		Methylergonovine	MERGV	0.049	0.161

**Table 2 toxins-18-00005-t002:** Prevalence and concentrations of 54 mycotoxins in European barley and wheat assessed over seven harvest years quantified by ultra-pressure liquid chromatography–tandem mass spectrometry.

	Barley					Wheat				
		Mycotoxin Concentration, µg/kg ^3^					Mycotoxin Concentration, µg/kg ^3^			
Mycotoxin ^1^	% > LOD ^2^	Median	Mean	Q3 ^2^	Max	% > LOD ^2^	Median	Mean	Q3 ^2^	Max
AFB1	1.27	0.00	0.02	0.00	2.17	0.27	0.00	0.01	0.00	2.98
AFB2	0.00	0.00	0.00	0.00	0.00	0.00	0.00	0.00	0.00	0.00
AFG1	0.00	0.00	0.00	0.00	0.00	0.13	0.00	0.00	0.00	0.60
AFG2	0.00	0.00	0.00	0.00	0.00	0.27	0.00	0.00	0.00	1.08
OTA	0.42	0.00	0.11	0.00	60.70	0.54	0.00	0.15	0.00	50.83
OTB	0.00	0.00	0.00	0.00	0.00	0.13	0.00	0.01	0.00	7.62
CIT	0.42	0.00	0.88	0.00	584.34	0.67	0.00	0.42	0.00	227.39
T2	27.46	0.00	6.23	3.94	267.41	9.02	0.00	0.68	0.00	76.61
HT2	33.24	0.00	16.54	16.78	370.62	13.86	0.00	2.80	0.00	86.12
DAS	0.28	0.00	0.01	0.00	6.78	0.00	0.00	0.00	0.00	0.00
NEO	2.82	0.00	0.47	0.00	52.72	0.54	0.00	0.12	0.00	41.24
DON	64.37	32.36	224.34	77.84	27,088.90	62.45	25.48	168.42	102.19	6758.43
AcDon3	13.66	0.00	8.58	0.00	1109.18	8.48	0.00	1.08	0.00	59.91
AcDon15	7.18	0.00	8.22	0.00	1011.43	5.92	0.00	1.31	0.00	89.57
DON3G	11.55	0.00	21.79	0.00	1463.31	8.88	0.00	6.16	0.00	394.51
NIV	0.00	0.00	0.00	0.00	0.00	0.13	0.00	0.40	0.00	295.70
FUSX	21.69	0.00	40.33	0.00	1000.68	16.42	0.00	10.59	0.00	1545.91
FB1	2.68	0.00	2.55	0.00	237.43	3.90	0.00	3.36	0.00	214.78
FB2	19.58	0.00	4.47	0.00	103.44	24.90	0.00	5.06	0.00	139.91
FB3	1.83	0.00	0.46	0.00	39.54	2.69	0.00	0.69	0.00	61.25
ZEA	5.49	0.00	6.16	0.00	924.65	5.38	0.00	2.06	0.00	307.83
FA	5.63	0.00	2.61	0.00	547.74	4.44	0.00	0.94	0.00	165.59
BEA	11.97	0.00	0.84	0.00	240.98	1.88	0.00	0.03	0.00	2.82
MON	31.83	0.00	15.96	9.10	627.89	14.27	0.00	2.32	0.00	163.80
ENN A/A1	75.07	9.75	44.85	41.16	843.03	65.01	2.94	8.69	7.37	955.47
ENN B/B1	75.07	38.79	243.54	186.78	4349.18	62.18	3.94	35.11	18.07	2246.33
PHOM A	0.14	0.00	0.06	0.00	39.41	0.00	0.00	0.00	0.00	0.00
AOL	0.00	0.00	1.93	0.00	235.94	0.00	0.00	1.84	0.00	180.35
PAT	0.42	0.00	0.36	0.00	110.80	0.54	0.00	1.76	0.00	517.41
MPA	5.35	0.00	2.91	0.00	350.95	1.35	0.00	0.79	0.00	206.47
ROQC	0.14	0.00	0.12	0.00	81.97	0.00	0.00	0.00	0.00	0.00
PA	0.56	0.00	0.19	0.00	40.17	0.00	0.00	0.00	0.00	0.00
CITR	1.69	0.00	1.41	0.00	290.63	0.67	0.00	0.35	0.00	153.02
WORT	0.00	0.00	0.00	0.00	0.00	0.27	0.00	0.07	0.00	38.99
GLIO	0.00	0.00	0.00	0.00	0.00	0.00	0.00	0.00	0.00	0.00
STMC	0.28	0.00	0.03	0.00	9.62	0.13	0.00	0.00	0.00	1.95
CPA	6.34	0.00	0.85	0.00	58.31	1.75	0.00	0.18	0.00	57.74
VERR	0.28	0.00	0.02	0.00	14.48	0.00	0.00	0.00	0.00	0.00
ERGMET	1.41	0.00	3.72	0.00	899.44	1.08	0.00	0.34	0.00	82.04
ERGAM	6.62	0.00	8.49	0.00	2303.68	4.98	0.00	3.70	0.00	1303.70
ERGCR	2.54	0.00	4.67	0.00	1772.95	2.15	0.00	1.61	0.00	364.94
ERGSI	4.08	0.00	2.17	0.00	529.27	2.96	0.00	1.05	0.00	220.62
ERGCO	3.24	0.00	4.71	0.00	2237.55	2.15	0.00	0.65	0.00	194.23
ERGCYP	3.66	0.00	6.16	0.00	1963.19	3.36	0.00	0.70	0.00	135.31
LYS	0.00	0.00	0.00	0.00	0.00	0.00	0.00	0.00	0.00	0.00
MERGV	0.56	0.00	0.00	0.00	1.23	0.00	0.00	0.00	0.00	0.00

^1^ Abbreviations for each mycotoxin are provided. For a detailed list of all abbreviations, see Table 1. ^2^ % > LOD: Percentage of samples with this mycotoxin greater than the limit of detection; Q3: third quartile; Max: maximum. ^3^ Median, mean, and quartile values were calculated using concentrations measured above the limit of quantification (LOQ).

**Table 3 toxins-18-00005-t003:** Prevalence and concentrations of 54 mycotoxins in European maize assessed over seven harvest years quantified by ultra-pressure liquid chromatography–tandem mass spectrometry.

	Maize				
		Mycotoxin Concentration, µg/kg ^3^			
Mycotoxin ^1^	% > LOD ^2^	Median	Mean	Q3 ^2^	Max
AFB1	12.66	0.00	4.48	0.00	451.33
AFB2	1.81	0.00	0.09	0.00	14.77
AFG1	3.10	0.00	0.14	0.00	15.67
AFG2	1.29	0.00	0.03	0.00	2.37
OTA	1.03	0.00	0.27	0.00	45.33
OTB	0.00	0.00	0.00	0.00	0.00
CIT	1.29	0.00	0.89	0.00	167.80
T2	8.79	0.00	3.14	0.00	272.31
HT2	4.91	0.00	3.20	0.00	414.51
DAS	1.55	0.00	0.08	0.00	9.12
NEO	2.58	0.00	0.34	0.00	24.43
DON	57.36	25.58	157.83	95.40	3517.18
AcDon3	8.53	0.00	1.84	0.00	136.97
AcDon15	25.58	0.00	16.96	6.36	614.42
DON3G	5.43	0.00	7.01	0.00	1138.91
NIV	1.03	0.00	3.10	0.00	409.02
FUSX	1.81	0.00	4.34	0.00	1249.24
FB1	76.23	532.28	1688.92	1638.27	25,004.39
FB2	79.84	36.33	108.93	92.48	3791.81
FB3	70.28	31.19	100.57	92.54	2139.50
ZEA	11.11	0.00	11.16	0.00	1198.24
FA	92.76	82.95	193.62	187.04	2050.33
BEA	23.00	0.00	6.01	0.00	413.46
MON	76.74	70.12	307.38	271.64	8825.47
ENN A/A1	31.78	0.00	1.76	4.02	13.69
ENN B/B1	40.57	0.00	2.89	2.56	156.13
PHOM A	0.26	0.00	0.01	0.00	5.67
AOL	0.00	0.00	0.46	0.00	55.14
PAT	0.00	0.00	0.00	0.00	0.00
MPA	5.17	0.00	3.14	0.00	484.25
ROQC	0.00	0.00	0.00	0.00	0.00
PA	1.81	0.00	1.05	0.00	89.20
CITR	0.00	0.00	0.00	0.00	0.00
WORT	0.00	0.00	0.00	0.00	0.00
GLIO	0.00	0.00	0.00	0.00	0.00
STMC	0.00	0.00	0.00	0.00	0.00
CPA	10.08	0.00	5.74	0.00	795.51
VERR	0.00	0.00	0.00	0.00	0.00
ERGMET	0.00	0.00	0.00	0.00	0.00
ERGAM	0.78	0.00	0.02	0.00	2.39
ERGCR	0.00	0.00	0.00	0.00	0.00
ERGSI	0.00	0.00	0.00	0.00	0.00
ERGCO	0.00	0.00	0.00	0.00	0.00
ERGCYP	0.00	0.00	0.00	0.00	0.00
LYS	0.00	0.00	0.00	0.00	0.00
MERGV	0.00	0.00	0.00	0.00	0.00

^1^ Abbreviations for each mycotoxin are provided. For a detailed list of all abbreviations, see Table 1. ^2^ % > LOD: Percentage of samples with this mycotoxin greater than the limit of detection; Q3: third quartile; Max: maximum. ^3^ Median, mean, and quartile values were calculated using concentrations measured above the limit of quantification (LOQ).

**Table 4 toxins-18-00005-t004:** Prevalence and concentrations of 54 mycotoxins in European maize silage and grass silage assessed over seven harvest years quantified by ultra-pressure liquid chromatography–tandem mass spectrometry.

	Maize Silage					Grass Silage				
		Mycotoxin Concentration, µg/kg ^3^					Mycotoxin Concentration, µg/kg ^3^			
Mycotoxin ^1^	% > LOD ^2^	Median	Mean	Q3 ^2^	Max	% > LOD ^2^	Median	Mean	Q3 ^2^	Max
AFB1	2.74	0.00	0.59	0.00	152.37	1.50	0.00	0.37	0.00	78.33
AFB2	0.91	0.00	1.13	0.00	161.94	0.64	0.00	0.21	0.00	67.77
AFG1	0.00	0.00	0.00	0.00	0.00	0.00	0.00	0.00	0.00	0.00
AFG2	0.00	0.00	0.00	0.00	0.00	1.07	0.00	2.08	0.00	293.30
OTA	0.00	0.00	0.00	0.00	0.00	0.00	0.00	0.00	0.00	0.00
OTB	0.00	0.00	0.00	0.00	0.00	0.00	0.00	0.00	0.00	0.00
CIT	0.61	0.00	1.73	0.00	383.63	0.21	0.00	0.36	0.00	169.42
T2	3.35	0.00	0.36	0.00	34.78	1.07	0.00	0.11	0.00	29.87
HT2	31.10	0.00	56.34	28.90	4230.30	2.36	0.00	1.42	0.00	207.90
DAS	0.30	0.00	0.01	0.00	2.45	0.86	0.00	0.07	0.00	10.31
NEO	1.52	0.00	8.23	0.00	1222.34	0.21	0.00	0.14	0.00	63.46
DON	96.65	569.42	1492.60	1487.56	72,361.90	48.61	0.00	39.52	58.21	648.72
AcDon3	20.73	0.00	5.65	0.00	269.22	0.21	0.00	0.08	0.00	36.33
AcDon15	18.90	0.00	69.30	0.00	5389.40	1.71	0.00	4.63	0.00	941.10
DON3G	1.22	0.00	6.39	0.00	1160.07	0.21	0.00	0.08	0.00	36.18
NIV	1.22	0.00	7.22	0.00	1069.79	0.64	0.00	1.36	0.00	236.60
FUSX	3.66	0.00	20.44	0.00	1380.82	9.21	0.00	73.67	0.00	2973.27
FB1	39.33	0.00	215.75	167.14	6879.26	4.07	0.00	4.16	0.00	306.57
FB2	42.99	0.00	40.41	39.19	888.97	11.13	0.00	4.01	0.00	233.84
FB3	31.71	0.00	16.86	17.70	354.36	3.85	0.00	0.92	0.00	77.52
ZEA	18.29	0.00	65.51	0.00	1911.70	2.78	0.00	4.18	0.00	491.51
FA	89.02	422.15	673.89	849.76	35,870.67	59.10	14.38	39.84	42.57	546.14
BEA	14.33	0.00	2.82	0.00	495.44	0.64	0.00	0.07	0.00	23.36
MON	39.94	0.00	16.81	15.30	308.84	8.14	0.00	0.94	0.00	34.84
ENN A/A1	54.88	2.87	10.24	13.27	143.98	37.04	0.00	8.78	8.75	516.03
ENN B/B1	55.79	4.31	28.79	19.32	1130.38	37.90	0.00	8.07	6.03	218.59
PHOM A	0.30	0.00	0.00	0.00	0.82	0.21	0.00	0.00	0.00	0.86
AOL	0.00	0.00	0.03	0.00	10.66	0.00	0.00	1.61	0.00	153.56
PAT	0.30	0.00	1.06	0.00	346.87	1.28	0.00	3.55	0.00	648.79
MPA	3.35	0.00	5.53	0.00	993.14	2.78	0.00	11.43	0.00	3293.50
ROQC	0.61	0.00	0.26	0.00	75.75	3.64	0.00	21.55	0.00	4655.03
PA	19.51	0.00	20.22	0.00	794.59	50.75	25.44	246.69	313.18	3945.90
CITR	0.00	0.00	0.00	0.00	0.00	0.00	0.00	0.00	0.00	0.00
WORT	0.00	0.00	0.00	0.00	0.00	0.00	0.00	0.00	0.00	0.00
GLIO	0.61	0.00	2.63	0.00	805.30	0.00	0.00	0.00	0.00	0.00
STMC	0.00	0.00	0.00	0.00	0.00	0.21	0.00	0.00	0.00	0.95
CPA	3.35	0.00	1.33	0.00	145.13	1.93	0.00	0.26	0.00	46.60
VERR	0.00	0.00	0.00	0.00	0.00	0.00	0.00	0.00	0.00	0.00
ERGMET	0.00	0.00	0.00	0.00	0.00	0.86	0.00	0.09	0.00	21.33
ERGAM	0.30	0.00	0.01	0.00	2.67	3.00	0.00	7.57	0.00	889.86
ERGCR	0.61	0.00	0.80	0.00	247.24	1.50	0.00	2.11	0.00	616.20
ERGSI	0.61	0.00	0.07	0.00	14.09	2.14	0.00	1.55	0.00	184.09
ERGCO	0.30	0.00	0.08	0.00	24.69	0.00	0.00	0.00	0.00	0.00
ERGCYP	0.30	0.00	0.35	0.00	115.32	1.71	0.00	1.03	0.00	235.78
LYS	0.00	0.00	0.00	0.00	0.00	0.00	0.00	0.00	0.00	0.00
MERGV	0.00	0.00	0.00	0.00	0.00	0.00	0.00	0.00	0.00	0.00

^1^ Abbreviations for each mycotoxin are provided. For a detailed list of all abbreviations, see Table 1. ^2^ % > LOD: Percentage of samples with this mycotoxin greater than the limit of detection; Q3: third quartile; Max: maximum. ^3^ Median, mean, and quartile values were calculated using concentrations measured above the limit of quantification (LOQ).

**Table 5 toxins-18-00005-t005:** Concentration (µg/kg) of type B trichothecenes by European climatic region in barley, maize, wheat, maize silage, and grass silage quantified by ultra-pressure liquid chromatography–tandem mass spectrometry assessed over seven harvest years.

	Mycotoxin Concentration, µg/kg ^1^						
Climatic Region	Mean	SD ^2^	Q1 ^2^	Median	Q3 ^2^	Maximum	*p* Value ^3^
	**Barley**						
Continental	341.41	1179.63	0.00	45.69	162.85	14,074.48	<0.005
Mediterranean	303.40	983.02	0.00	70.13	201.19	6459.48	
Nordic	3783.15	6997.05	480.84	1476.99	3428.30	28,987.66	
Oceanic	119.94	195.30	0.00	49.64	154.16	1853.14	
	**Maize**						
Continental	89.08	187.81	0.00	17.04	71.20	1023.93	<0.005
Mediterranean	166.34	471.74	0.00	29.04	98.06	3974.41	
Nordic							
Oceanic	868.07	777.73	264.66	798.56	1181.07	2832.32	
	**Wheat**						
Continental	262.14	733.94	0.00	33.50	149.49	7281.83	<0.005
Mediterranean	228.61	504.72	0.00	24.52	170.41	2506.77	
Nordic							
Oceanic	110.66	259.48	0.00	29.66	92.28	2382.64	
	**Maize Silage**						
Continental	2169.62	7351.81	329.82	799.18	1808.24	79,024.12	0.089
Mediterranean	397.82	704.50	106.99	204.28	446.43	4903.30	
Nordic							
Oceanic	1281.09	1672.37	331.00	643.82	1705.34	11,374.18	
	**Grass Silage**						
Continental	108.88	320.58	0.00	7.46	71.48	2620.84	0.857
Mediterranean	0.00		0.00	0.00	0.00	0.00	
Nordic	84.23	69.60	34.68	60.29	129.06	217.97	
Oceanic	127.70	300.49	0.00	35.01	91.40	2973.27	

^1^ B trichothecenes are represented by the sum of deoxynivalenol, 3-acetyl-deoxynivalenol, 15-acetyl-deoxynivalenol, deoxynivalenol-3-glucoside, nivalenol, and fusarenon X. Median, mean, and quartile values were calculated using concentrations measured above the limit of quantification (LOQ). ^2^ SD: Standard deviation; Q1: first quartile; Q3: third quartile. ^3^ *p*-value for the mean for climatic region effect within each feedstuff.

**Table 6 toxins-18-00005-t006:** Concentration (µg/kg) of emerging mycotoxins by European climatic region in barley, maize, wheat, maize silage, and grass silage quantified by ultra-pressure liquid chromatography–tandem mass spectrometry assessed over seven harvest years.

	Mycotoxin Concentration, µg/kg ^1^						
Climatic Region	Mean	SD ^2^	Q1 ^2^	Median	Q3 ^2^	Maximum	*p* Value ^3^
	**Barley**						
Continental	163.95	458.51	1.02	17.88	90.32	3247.41	<0.005
Mediterranean	59.91	173.60	3.22	12.43	29.49	1036.00	
Nordic	1560.89	1439.75	692.51	1279.78	1754.40	5144.72	
Oceanic	374.81	679.76	25.57	107.16	410.81	4611.46	
	**Maize**						
Continental	283.91	697.50	15.00	64.08	174.18	4750.76	0.316
Mediterranean	346.99	835.37	20.27	97.77	314.90	8844.40	
Nordic							
Oceanic	100.95	154.61	5.37	35.60	134.34	507.68	
	**Wheat**						
Continental	27.66	70.11	1.22	7.08	19.85	876.73	<0.005
Mediterranean	14.94	44.12	0.00	3.24	14.01	340.54	
Nordic							
Oceanic	67.41	231.29	5.27	14.95	45.18	3114.48	
	**Maize Silage**						
Continental	53.38	78.64	4.69	25.48	71.18	538.86	0.078
Mediterranean	39.66	87.57	0.00	3.11	29.04	495.44	
Nordic							
Oceanic	77.15	149.64	1.04	30.29	80.51	1197.93	
	**Grass Silage**						
Continental	33.31	62.29	0.00	10.59	38.12	526.16	<0.005
Mediterranean	129.11		129.11	129.11	129.11	129.11	
Nordic	28.69	40.04	9.27	16.85	30.84	164.85	
Oceanic	10.55	27.96	0.00	0.00	9.86	214.45	

^1^ Emerging mycotoxins are represented by the sum of alternariol, beauvericin, enniatin A/A1, enniatin B/B1, moniliformin, and phomopsin A. Median, mean, and quartile values were calculated using concentrations measured above the limit of quantification (LOQ). ^2^ SD: Standard deviation; Q1: first quartile; Q3: third quartile. ^3^ *p*-value for the mean for climatic region effect within each feedstuff.

**Table 7 toxins-18-00005-t007:** Probability of co-occurrence between pairs of mycotoxin groups for barley, maize, and wheat grains. Pairs with significant (*p* < 0.05) positive (orange) or negative (blue) co-occurrences based on model expectations are shown, along with those pairs having random (gray) co-occurrences. Pairs that showed no co-occurrence are blank (white).

	Group ^1^									
Group ^1^	OCH/CIT	Type B	Type A	FUMs	ZEA	FA	Emerg.	Pen.	Asp.	Ergot
	**Barley**									
AFs		0.009	0.005	0.003			0.011			
OCH/CIT		0.006	0.003	0.002			0.007			
B Tricho.			0.279	0.151	0.041	0.040	0.609	0.054	0.048	0.062
A Tricho.				0.082	0.022	0.022	0.330	0.029	0.026	0.033
FUMs					0.012	0.012	0.179	0.016	0.014	0.018
ZEA						0.003	0.049	0.004	0.004	0.005
FA							0.048	0.004	0.004	0.005
Emerg.								0.063	0.057	0.073
Pen.									0.005	0.006
Asp.										0.057
	**Maize**									
AFs	0.004	0.082	0.017	0.111	0.016	0.127	0.119	0.010	0.014	
OCH/CIT		0.017	0.003	0.023	0.003	0.026	0.025		0.003	
B Tricho.			0.073	0.490	0.068	0.559	0.524	0.042	0.061	0.005
A Tricho.				0.099	0.014	0.113	0.106	0.008	0.012	
FUMs					0.093	0.755	0.709	0.057	0.082	0.006
ZEA						0.105	0.099	0.008	0.011	
FA							0.808	0.065	0.093	0.007
Emerg.								0.061	0.088	0.007
Pen.									0.007	
Asp.										
	**Wheat**									
AFs										
OCH/CIT										
B Tricho.			0.114	0.185	0.041	0.029	0.529			0.043
A Tricho.				0.049	0.011	0.008	0.140			0.011
FUMs					0.018	0.013	0.229			0.019
ZEA						0.003	0.051			0.004
FA							0.036			0.003
Emerg.										0.053
Pen.										
Asp.										

^1^ AFs: Total aflatoxins; OCH/CIT: ochratoxins/citrinin; B Tricho.: type B trichothecenes; A Tricho.: type A trichothecenes; FUMs: total fumonisins; ZEA: zearalenone; FA: fusaric acid; Emerg.: emerging mycotoxins; Pen.: *Penicillium* mycotoxins; Asp.: *Aspergillus* mycotoxins; Ergot: ergot alkaloids.

**Table 8 toxins-18-00005-t008:** Probability of co-occurrence between pairs of mycotoxin groups for maize silage and grass silage. Pairs with significant (*p* < 0.05) positive (orange) or negative (blue) co-occurrences based on model expectations are shown, along with those pairs having random (gray) co-occurrences. Pairs that showed no co-occurrence are blank (white).

	Group ^1^									
Group ^1^	OCH/CIT	Type B	Type A	FUMs	ZEA	FA	Emerg.	Pen.	Asp.	Ergot
	**Maize Silage**									
AFs		0.033	0.011	0.016	0.006	0.030	0.026	0.008		
OCH/CIT		0.006				0.005	0.005			
B Tricho.			0.322	0.455	0.177	0.863	0.742	0.222	0.041	0.012
A Tricho.				0.156	0.061	0.296	0.254	0.076	0.014	0.004
FUMs					0.086	0.418	0.359	0.107	0.020	0.006
ZEA						0.163	0.140	0.042	0.008	
FA							0.681	0.204	0.038	0.011
Emerg.								0.175	0.033	0.009
Pen.									0.010	
Asp.										** 0.057 **
	**Grass Silage**									
AFs		0.012		0.003		0.013	0.011	0.012		
OCH/CIT										
B Tricho.			0.022	0.076	0.015	0.318	0.288	0.292	0.012	0.030
A Tricho.				0.006		0.024	0.022	0.022		0.002
FUMs					0.004	0.084	0.076	0.077	0.003	0.008
ZEA						0.016	0.015	0.015		
FA							0.316	0.321	0.013	0.033
Emerg.								0.291	0.011	0.030
Pen.									0.012	0.030
Asp.										

^1^ AFs: Total aflatoxins; OCH/CIT: ochratoxins/citrinin; B Tricho.: type B trichothecenes; A Tricho.: type A trichothecenes; FUMs: total fumonisins; ZEA: zearalenone; FA: fusaric acid; Emerg.: emerging mycotoxins; Pen.: *Penicillium* mycotoxins; Asp.: *Aspergillus* mycotoxins; Ergot: ergot alkaloids.

**Table 9 toxins-18-00005-t009:** Number of samples utilized in this survey by feedstuff type, European climatic region, and harvest year.

Region	Barley	Maize	Wheat	Maize Silage	Grass Silage
Total sample number	710	387	743	328	467
	**Region**				
Continental	254	85	321	171	166
Mediterranean	45	279	69	53	1
Nordic	18	0	0	0	15
Oceanic	393	23	353	104	285
	**Harvest year**				
2018	38	15	46	12	10
2019	30	17	45	8	17
2020	71	49	79	49	22
2021	179	102	177	79	70
2020	113	84	94	50	103
2023	143	82	178	101	132
2024	136	38	124	29	113

## Data Availability

The original contributions presented in this study are included in the article/Appendix A. Further inquiries can be directed to the corresponding author.

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
