# Peer review of "Influence of Climatic Region and Feedstuff Type on the Co-Occurrence and Contamination Profiles of 54 Mycotoxins in European Grains and Forages: A Seven-Year Survey"

_toxins, 2025, doi:10.3390/toxins18010005_

Round 1
Reviewer 1 Report
Comments and Suggestions for Authors
The manuscript provides novel insights into mycotoxin occurrence and co-occurrence across multiple European regions. The seven-year survey, encompassing a large number of samples stratified by feedstuff, region, and harvest year, offers valuable information for monitoring mycotoxin risk over time and informing mitigation strategies. The regional differentiation and co-occurrence patterns are particularly noteworthy, even though they are not linked to climatic factors or fungal community composition, an understandable limitation given the scope and cost of analysing such an extensive dataset. The detection of marked regional differences, especially the elevated risks associated with barley from Nordic countries and maize silage, is a relevant contribution. The inclusion of emerging mycotoxins and the assessment of 54 analytes further strengthen the study. Overall, the results, statistical analyses, and figures are solid, and the discussion is well aligned with the findings. The use of the “coccur” R package to calculate co-occurrence probabilities is well established and very helpful to readers in understanding the correlation matrix.
For future studies, exploring links between key climatic variables and regional mycotoxin patterns could be valuable, although beyond the current scope. A second point concerns the silage samples: farm-level conditions vary widely, limiting comparability across years. The authors should therefore emphasise the need for caution regarding the traceability and variability of silage data in longitudinal surveys. Additionally, given the growing relevance and emerging risks associated with Alternaria toxins, it would have been valuable to include a broader set of metabolites beyond alternariol, such as tenuazonic acid, alternariol monomethyl ether, and altenuene.
I have only a few minor suggestions:
-Line 35: Please add “, among others” at the end of the sentence. Because numerous fungal genera can produce mycotoxins beyond those listed.
-Lines 45-50: Please add some references to support these statements.
-Lines 51-57: Please, add some statements regarding the state of the art in other agricultural regions such as Latin America, Africa, Asia, etc. Comparing only the US with Europe could bias the study approach. Moreover, these could help to complement the introduction section, which, in my opinion, is concise but short.
-Line 77: I suggest changing the title section. I consider the term “presence” as inappropriate. Please replace with “Mycotoxin contamination” or a similar term.
-Line 81: Please specify the composition of grass silage. Which grass species? Grass silage can vary widely across agricultural regions, as can the associated fungal community.
-Line 377: Please correct the fungal genus name (“Claviceps” is the correct form).
-Tables: Please fit all the tables to the page margins for a tidier presentation.
For all the reasons mentioned previously, I consider that the current version of the manuscript must address these suggested changes (minor revisions) before publication in Toxins.
Reviewer 2 Report
Comments and Suggestions for Authors
The manuscript provides a robust and comprehensive dataset, offering valuable insights into mycotoxin occurrence patterns across different feedstuff types, climatic regions, and harvest years. Its major strengths include the large sample size (2,635 samples), the use of a validated analytical method (UPLC-MS/MS), and detailed statistical evaluation of co-occurrence patterns. These elements significantly advance understanding of multi-mycotoxin contamination risks in animal feed and support improved risk assessment and feed safety management.
- The introduction should better highlight how this study advances beyond previous European surveys. Emphasize unique contributions such as climatic region analysis and probabilistic co-occurrence modeling.
- The discussion on climate impact is largely speculative. Consider integrating actual weather data (temperature, precipitation) into the analysis or clearly state this as a limitation.
- Figures 1–6 are informative but visually dense. Improve readability by adding clearer legends and highlighting key findings (e.g., regions with highest contamination).
- Tables 1–5 contain extensive numeric data; summarizing trends in text or using heatmaps would make interpretation easier.
- While the probabilistic approach is well-described, the biological implications of positive versus negative associations need deeper discussion. For example, explain why grass silage shows more negative associations compared to grains.
- Strengthen the conclusion by explicitly linking findings to feed safety management strategies and regulatory frameworks. Provide actionable recommendations for producers
- Ensure references are up-to-date and formatted according to journal guidelines.
- Summarize key supplementary findings in the main text for better accessibility.
The manuscript is scientifically sound and provides significant data for risk assessment. With improved contextualization, clearer data visualization, and stronger linkage to practical applications, it will make a valuable contribution to the field.
Comments on the Quality of English Language
- Revise long sentences in the discussion for clarity and correct minor grammatical issues (e.g., simplify “gauged significance at an alpha value”).
Round 2
Reviewer 2 Report
Comments and Suggestions for Authors
The authors have provided all the changes requested in the initial review. The manuscript now presents a clear and well-structured analysis of the occurrence, concentration and co-occurrence patterns of mycotoxins in different feed types and climatic regions in Europe.
Comments on the Quality of English Language
- Revise long sentences in the discussion for clarity and correct minor grammatical issues (e.g., simplify “gauged significance at an alpha value”).